# Highlighting Curcumin-Induced Crosstalk between Autophagy and Apoptosis as Supported by Its Specific Subcellular Localization

**DOI:** 10.3390/cells9020361

**Published:** 2020-02-04

**Authors:** Francisco J. Sala de Oyanguren, Nathan E. Rainey, Aoula Moustapha, Ana Saric, Franck Sureau, José-Enrique O’Connor, Patrice X. Petit

**Affiliations:** 1Laboratory of Cytomics, Joint Research Unit, University of Valencia, Avda. Blasco Ibanez 15, 46010 Valencia and Principe Felipe Research Center, Cerrer d’Eduardo Primo Yufera 3, 46012 Valencia, Spain; fransalade@gmail.com (F.J.S.d.O.); jose.e.oconnor@uv.es (J.-E.O.); 2CNRS UMR 8003, SPPIN—Saints Pères Paris Institute for the Neurosciences, Univerity of Paris, Campus Saint-Germain, 45 rue des Saint-Pères, 75006 Paris, France; nathan.rainey@inserm.fr (N.E.R.); aoulamou@yahoo.com (A.M.); ana.saric2404@gmail.com (A.S.); 3LVTS—Laboratory for Vascular Translational Science, UMR1148 INSERM, Université Paris XIII, Sorbonne Paris Cité, F-93017 Bobigny, France; 4Division of Molecular Medicine, Ruder Boškovic Institute, Bijenička cesta 54, 10000 Zagreb, Croat; 5Université Sorbonne Paris Cité—UPMC, Paris 6, Laboratoire Jean Perrin, 75005 Paris, France; franck.sureau@upmc.fr

**Keywords:** apoptosis, autophagy, calcium, cancer, cell death, endoplasmic reticulum, lysosome, real-time cellular impedance, ROS, xCELLigence

## Abstract

Curcumin, a major active component of turmeric (*Curcuma longa*, L.), is known to have various effects on both healthy and cancerous tissues. In vitro studies suggest that curcumin inhibits cancer cell growth by activating apoptosis, but the mechanism underlying the anticancer effect of curcumin is still unclear. Since there is a recent consensus about endoplasmic reticulum (ER) stress being involved in the cytotoxicity of natural compounds, we have investigated using Image flow cytometry the mechanistic aspects of curcumin’s destabilization of the ER, but also the status of the lysosomal compartment. Curcumin induces ER stress, thereby causing an unfolded protein response and calcium release, which destabilizes the mitochondrial compartment and induce apoptosis. These events are also associated with secondary lysosomal membrane permeabilization that occurs later together with an activation of caspase-8, mediated by cathepsins and calpains that ended in the disruption of mitochondrial homeostasis. These two pathways of different intensities and momentum converge towards an amplification of cell death. In the present study, curcumin-induced autophagy failed to rescue all cells that underwent type II cell death following initial autophagic processes. However, a small number of cells were rescued (successful autophagy) to give rise to a novel proliferation phase.

## 1. Introduction

Curcumin, a major bioactive compound in turmeric, has a broad spectrum of activity, including antioxidant, anticarcinogenic, and anti-inflammatory properties [1,2,3]—Curcumin has a symmetric molecular organization and is defined as a diferuloyl methane [4]. Curcumin has the following chemical formula: (1E,6E)-1,7-bis(4-hydroxy-3-methoxyphenyl)-1,6-heptadiene-3,5-dione (C_21_H_20_O_6_). Its structure contains three chemical entities: two aromatic ring systems with an o-methoxy phenolic group, connected by a seven-carbon linker consisting of an α,β-unsaturated β-diketone moiety. Double bonds inside the molecule account for its participation in many electron transfer reactions. Curcumin is an electron donor and stabilizes its chemical structure by redistribution and resonance of the electron cloud [4]. Curcumin presents UV-visible absorption bands (250–270 nm and 350–450 nm, respectively), so curcumin fluoresces with a maximum emission at 470 nm. These optical properties have been used to isolate curcumin using various techniques, such as high-performance liquid chromatography (HPLC). Its fluorescence enables us to follow very low amounts of curcumin and its related metabolites in plasma and urine at concentrations as low as 2.5 ng/mL [5,6,7]. Curcumin can also be excited at 488 nm, with a lower fluorescent yield emission in the 500–530 nm range, for detection by flow cytometry and confocal microscopy. Curiously, this has rarely been used for curcumin imaging at the cellular level [8,9]. Curcumin is a hydrophobic molecule with a log P value of 3.0 at neutral pH [10]. Therefore, curcumin is not easily soluble in physiological media and exhibits poor distribution and bioavailability [11].

The main chemical feature of the curcumin molecule is the presence of o-methoxyphenol group and methylenic hydrogen responsible for the donation of an electron/hydrogen to reactive oxygen species, hence eliciting antioxidant activity. Curcumin is thought to be as efficient in the removal of oxyradicals as well-known antioxidants-thiols, vitamin A, vitamin C and vitamin E—and mimics the function of a superoxide dismutase [10].

The unsaturated ketone of curcumin undergoes a nucleophilic addition reaction as an acceptor, together with A-OH, A-SH, and A-SeH that sustain its interaction with multiple molecules [10]. Curcumin has also been suggested to change the properties of cell membranes in which its inserts and indirectly affect membrane-bound proteins [12,13]. The interaction of curcumin with artificial membrane (DOPC-based membranes) shows that it thinned the bilayer and weakened its elasticity modulus [14,15]. As a result, curcumin can insert in proteo-lipidic compartments of the cells and/or binds covalently to various proteins in the cytosol, thus influencing protein functions at the organelle membrane surfaces [16]. As an exemple, the conjugation of curcumin with thiols results in the depletion of glutathione and impairs association with the cell antioxidant defense system. In this regard, the depletion of glutathione suggests that curcumin could act as a pro-oxidant in some conditions [17].

At low concentrations, curcumin reacts as an antioxidant, but acquires pro-oxidant properties above 20 μM, thus revealing its hormetic behavior [8,18]. Curcumin is a pleiotropic molecule which interacts with multiple targets involved in inflammatory reactions, such as tumor necrosis factor-alpha (TNFα) and interleukins (ILs) [19].

Curcumin also interacts with a number of biomolecules through non-covalent and covalent binding. The hydrogen bonding and hydrophobicity of curcumin, arising from its aromatic and tautomeric structures along with the flexibility of the linker group, are responsible for the noncovalent interactions. Curcumin directly interacts with numerous proteins in the cytoplasm and this may explain its pleotropic and multiple intracellular effects [20,21].

In the present study, we investigated in more detail the intracellular localization of curcumin, since all curcumin-induced cell signaling depends on its precise insertion into well-defined intracellular membranes. Here, we confirm that curcumin mainly targets the endoplasmic reticulum (ER) and is associated with ER swelling and a combination of an unfolded protein response (UPR) response and calcium release. We point out a secondary targeting of curcumin at the lysosomal pathway, which potentiates the induction of cell death.

We investigated the mechanism whereby curcumin modulates oxidative stress-mediated signaling of inflammatory responses leading to autophagy. By using impedancemetry (xCELLigence system), we depicted for the first time a situation where curcumin at two different concentrations induces, on the one hand—successful autophagy, and on the other hand—classic apoptotic cell death. The integration impedancemetry and flow cytometry helped us better understand the molecular mechanisms of action of curcumin. Detailed knowledge of curcumin-induced autophagy and possible type II cell death and the relationship between the two pathways may lead to the pharmacological development of novel therapeutic approaches, for example, to liver tumorigenesis and neurological disease.

## 2. Materials and Methods

### 2.1. Chemicals and Reagents

Calcein-AM, curcumin, propidium iodide (PI), *N*-acetylcysteine (NAC) and ruthenium red were from Sigma–Aldrich Chemical Co. (St. Louis, MO, USA). Culture medium RPMI-1640, fetal bovine serum (FBS), penicillin-streptomycin and L-glutamine were from GIBCO BRL (Invitrogen, Grand Island, NY, USA). Calcein-AM, 2,7-dichlorodihydrofluorescein diacetate (DCFH-DA), 3,3′- dihexyloxacarbocyanine iodide [DiOC_6_(3)] and *N*-[4-[6-[(acetyloxy)methoxy]-2,7-dichloro-3-oxo-3H-xanthen-9-yl]-2-[2-[2-[*bis*[2[(acetyloxy)methoxy]-2-oxyethyl]amino]-5-methyl-phenoxy]ethoxy]phenyl-*N*-[2-[(acetyloxy)methoxy]-2-oxyethyl]-(acetyloxy)methyl ester (Fluo-4/AM) were from Molecular Probes (Invitrogen, Eugene, OR, USA). M. Murphy (Medical Research Council Dunn Human Nutrition Unit, Wellcome Trust/MRC Building, Hills Road, Cambridge CB2 2XY, United Kingdom) and VP Skulachev (Belozersky Institute of Physico-Chemical Biology, Lomonosov Moscow State University, Vorob’evy Gory 1/40, 119,992 Moscow, Russia) provided us with chemical products and advice concerning the mitochondrially targeted anti-oxidants MitoQ_10_ and SKQ1, respectively. Acridine orange (AO) (Molecular Probes, Eugene, OR, cat. no. A1301) stock solution was made in water (1 mg/mL) and stored at 4 °C, and LysoTracker Red DN 99 (LyR) (Molecular Probes, Eugene, OR, cat. no. L7528) was dissolved in PBS (50 nM) and stored at 4 °C. Aliquots of the stock solutions of the dyes were added directly to culture media. Prior to imaging, cells were incubated for 15 min with the dye at 4 μM (AO) or 100 nM (LyR).

### 2.2. Cells

Human hepatoma-derived Huh-7 cells were from the RIKEN BioResource Center, Tsukuba, Japan and were grown in the presence of 5% CO_2_ with Dulbecco’s modified Eagle’s medium (DMEM) containing high glucose (25 mM Sigma–Aldrich, St. Louis, MI, USA) with 10% FBS (Hyclone, Logan, UT, USA) completed with 1% penicillin-streptomycin, HEPES NaOH 1 mM, Na-pyruvate 1 mM and 1% non-essential amino acids (MEAM, GIBCO).

### 2.3. Microspectrofluorometry

The UV-visible confocal laser microspectrofluorometer prototype was built around a Zeiss UMSP 80 UV epifluorescence microscope (Carl Zeiss, Inc., Oberkochen, Germany), optically coupled by UV reflecting mirrors to a Jobin-Yvon HR640 spectrograph (ISA, Longjumeau, France) [22]. The 351 nm UV line of an argon laser (model 2025; Spectra-Physics, Mountain View, CA, USA) was used for either drug or fluorochrome excitation. The diameter of the laser beam was first enhanced through a double-lens beam expander in order to cover the entire numerical aperture of the microscope’s optics. The laser beam was then deflected by an epi-illumination system (dichroic mirror or semireflecting glass) and focused on the sample through the microscope objective (X63 Zeiss Neofluar water-immersion objective; numerical aperture = 1.2) on a circular spot 0.5 µm in diameter. The excitation power was reduced to less than 0.1 µW by neutral density optical filters. The objective was immersed in the culture medium, and a circular area 0.8 µm in diameter was selected at the sample level, by interposing a field diaphragm in the emission pathway of the microscope, to selectively collect the fluorescence signal from the nucleus or a specific cytoplasmic area. Confocal conditions are met when the image of this analysis field diaphragm through the microscope objective perfectly coincides with the focus of the laser beam on the sample.

Under these conditions, the experimental spatial resolution, measured on calibrated latex beads (2 µm, 0.6 µm, and 0.16 µm in diameter) labeled with the fluorescent probe fluorescein, was 0.5 µm for the directions X, Y, and Z. Finally, the fluorescence spectra were recorded after spectrograph dispersion, in the 380–630 nm region on a 1024 diode-intensified optical multichannel analyzer (Princeton Instruments, Inc., Princeton, NJ, USA) with a resolution of 0.25 nm/diode. Each fluorescence emission spectrum was collected from 1 to 10 s. Data were stored and processed on an 80,286 IBM PS/2 microcomputer using the Jobin-Yvon “Enhanced Prism” software. It should be noted that, in order to avoid any possible fluorescence from a plastic or glass support during analysis with near-UV excitation, cells were grown on quartz plates that were then placed on the microscope stage in 50 mm thermostated Petri dishes, filled with 5 mL of phosphate-buffered saline (PBS). A uranyl glass bar was used as a fluorescent standard to control laser power and instrumental response, and to enable quantitative comparison between spectra recorded on different days. Sample heating, photobleaching, and photo damage were assessed empirically and found to be negligible under our experimental conditions. In particular, cells always remained viable after repeated fluorescence determinations, as controlled by phase-contrast microscopy.

### 2.4. Imaging Flow Cytometry

Imaging flow cytometry combines the strengths of microscopy and flow cytometry in a single system for quantitative image-based cellular assays in large and heterogeneous cell populations. The Amnis ImageStream100 instrument (Amnis, Merck Millipore, Burlington, MA, USA) acquires up to six images of each cell in different imaging modes. The system is equipped with 3 lasers (405 nm, 488 nm and 640 nm) and cells can be magnified by a 20, 40 or 60 × objective, allowing a wide range of applications. The six images of each cell comprise: a side-scatter (SSC) image, a transmitted light (brightfield) image and four fluorescence images corresponding to the FL1, FL2, FL3 and FL4 spectral bands of conventional flow cytometers [23].

Light is quantified for all the pixels in the image and identifies both the intensity and location of the fluorescence. Each pixel has an X coordinate, a Y coordinate, and an intensity value that corresponds to the amount of light captured at that location. Validation and quality control of the cytometer were performed by daily running of SpeedBeads™ reagent (Amnis, Merck Millipore) and using INSPIRE™ 2.2 software (Amnis). Data analysis was performed using IDEAS 6.0 software™ (Amnis) and single-color controls were used to create a compensation matrix that was applied to all files to correct for spectral crosstalk.

### 2.5. Determination of Mitochondrial Membrane Potential (∆Ψm), Reactive Oxygen Species, and Cytosolic Ca^2+^ Levels

A density of 2 × 10^6^ Huh-7 cells on 6-well plates was maintained with 25 μM curcumin for a given period of time ranging from 0 to 48 h depending on the experiments. After treatment, cells were trypsinized and then harvested, washed, and resuspended together with their supernatant in PBS. 3,3′-Dihexyloxacarbocyanineiodide [DiOC_6_(3)] was added at 40 nM final concentration for ΔΨm determination, 2′,7′-dichlorodihydrofluorescein diacetate (DCFH-DA) at 5 μM for peroxidase activity, and MitoSOX at 1 μM for superoxide anion. Most of the time double staining was done in order to assay simultaneously cell viability, with propidium iodide (PI, stock solution, 1 mg.mL^−1^) for DiOC_6_(3), DCFH-DA and Fluo4-AM and with TO-PRO-3 iodide (stock solution, 1 mg.mL^−1^) for MitoSOX. A supplemental double staining was used for the distinction between viable, apoptotic, and necrotic cells with YO-PRO-1/PI (Molecular Probes) in parallel with annexin-V/PI staining done with annexin-V-FITC when needed (Immunotech, Beckman-Coulter) in the presence of calcium in order to detect the aberrant exposure of phophatidyl serine residues at the outer surface of the plasma membrane. All samples were analyzed using flow cytometry as previously described [24,25] on a FACS Calibur 4C.

### 2.6. Cell Cycle Analysis by Flow Cytometry

Huh-7 cells were taken from the xCELLigence wells at defined times along the different proliferation curves and their position in the cell cycle was evaluated by measuring 5-bromo-2-deoxyuridine (BrdU) incorporation using the APC BrdU Flow kit (catalogue number 552598, BD Pharmingen). Cells were incubated with BrdU (10 mM) for 1 h at 37 °C, washed, trypsinized, and fixed with cytofix/cytoperm buffer. BrdU staining was done following the kit procedure. DNA was stained with 7-aminoactinomycin D (7-AAD) and cells were analyzed using FACS Calibur4C (Becton Dickinson) with the FL-1 channel (530 ± 30 nm band pass) for BrdU and the FL-3 channel (670 nm long pass) for 7-ADD. The sub-G_0_G_1_ peak represented the dead cells within the samples.

### 2.7. Control of ROS Production by Antioxidants

To test the effects of ROS scavengers, N-acetylcysteine (NAC; 1 mM), MitoQ10 (5 μM) and SkQ1 (500 nM) cells were preincubated for 3 h with the antioxidants before treatments. Cell viability (PI) together with ΔΨm [DiOC_6_(3)] was measured.

### 2.8. Control of Calcium Efflux by Calcium Chelators or Inhibition of the Mitochondrial Calcium Uniport

For cytosolic calcium determination, Fluo4-AM (1 mM stock solution, Molecular Probes) was used at 5 μM final concentration, as previously described [25,26]. Ca^2+^-titration of Fluo4-AM was performed by reciprocal dilution of Ca^2+^-free and Ca^2+^-saturated buffers prepared using a Ca^2+^ kit buffer (Invitrogen) on cells poisoned with DPBS (phophate buffersaline) supplemented with carbonyl cyanide m-chlorophenylhydrazone (mClCCCP, 10 μM) and ionomycin (5 μM) [27]. Most of the time, double staining was performed to simultaneously assay cell viability with propidium iodide (PI) at 1 µg.mL^−1^ final concentration. Control analysis was done with curcumin at different concentrations and was found to be negligible when Fluo4-AM was added (since the fluorescence of the probes is substantial compared with the fluorescence of curcumin at the chosen setting).

### 2.9. Analysis of Cathepsin and Calpain Activities by Flow Cytometry

Cathepsin and calpain activities in live cells were determined with the use of specific substrates as previously described in [8]. 

### 2.10. Analysis of CHOP and GRP78 Activities by Flow Cytometry

Huh-7 cells were treated with curcumin at different times and tested for GRP78 and CHOP abundance. Cells were grown in 6-well plates, trypsinized and then fixed with 4% paraformaldehyde at 4 °C for 40 min and rinsed several times with PBS. Nonspecific binding sites were blocked for 2 h at room temperature with 5% normal SVF (Gibco, ThermoFisher Scientific, Waltham, MA, USA) in 0.1% Triton X-100-PBS. Caco-2 cells were incubated overnight at 4 °C with primary antibodies (1:100 dilutions with blocking buffer) for GRP78 (Cell Signaling, Danvers, MA, USA) or CHOP (Santa Cruz, CA, USA). Cells were then incubated with the appropriate fluorescein isothiocyanate or tetramethylrhodamine isothiocyanate-conjugated secondary antibodies (BD Biosciences, Grenoble, France) for 2 h at 4 °C. Cells were analyzed by flow cytometry using the green (FL1 = 530 ± 30 nM) or red (FL-2 = 585 ± 42 nm) channels. Each experiment was repeated at least four times in duplicate.

### 2.11. Caspase Activation, Fluorimetric Assays

Isolated Huh-7 cells were washed and suspended in calcium-free buffer solution (140 mM NaCl, 1.13 mM MgCl_2_, 4.7 mM KCl, 10 mM glucose, 0.1 M EDTA, and 10 mM HEPES, pH 7.2). Cells were then loaded at room temperature for 30 min with fluorescent indicator-linked substrates for activated caspase-8 (10 μM Z-IETD-R110; Molecular Probes), caspase-9 (10 μM Z-LEHD-R110; Molecular Probes), or caspases 3/7 (Caspase-3/7 Green ReadyProbes™ Reagent with a DEVD sequence, Molecular Probes).

### 2.12. Impedancemetry with xCELLigence for the Measurement of Cell Proliferation

All impedance measurements were performed with the xCELLigence RTCA MP instrument (ACEA Biosciences, San Diego, CA, USA). First, 50 µL of Huh-7 cell culture medium was added to each well of 16-well E-Plates (ACEA Biosciences) and the background impedance was measured and displayed as cell index [28,29]. In parallel, the xCELLigence system allowed us to choose a specific timeframe according to the impedance curve, detach quadruplicate wells, pool them, and run flow cytometry on cells of interest.

### 2.13. Measurement of Ca^2+^-ATPase Activity

Sarco-endoplasmic reticulum ATPase (SERCA) activity was measured with a Ca^2+^-ATPase assay kit according to the manufacturer’s protocols (Abcam, Cambridge, UK). Inorganic phosphate (Pi), which was generated from hydrolysis of ATP by ATPase, can be measured by a simple colorimetric reaction. Ca^2+^-ATPase activities were expressed in units of micromolar inorganic phosphate produced (mM Pi/mg protein/h).

### 2.14. Electron Microscopy

Huh-7 cells were fixed in 1.25% glutaraldehyde buffered with 0.1 M sodium phosphate (pH 7.4) for 24 h at 4 °C, dehydrated with ethanol at 4 °C, and immersed in a 1:1 mixture of propylene oxide and Epon. They were embedded in Epon by polymerization at 60 °C for 48 h and examined under the electron microscope.

## 3. Results

### 3.1. Uptake of Curcumin and Invalidation of Its Putative Cytoplasmic Membrane/Nuclei Localization

We previously demonstrated that curcumin enters cells quite rapidly, within seconds, to a final ratio between external added curcumin and the intracellular concentration of 1/20, that correspond to 1 to 2.5 μM as intracellular concentration for 20 and 50 μM added externally [8]. There is also a linear relationship between external and internal concentrations of curcumin in the range of 0 to 80 μM [8].

In the present study, we studied intracellular curcumin with two techniques: flow imaging (image flow cytometry, Amnis technology) and flow cytometry (FACS calibur 4C). First, image cytometry confirmed rapid cell staining by 5 µM curcumin for 3 h (Figure 1a). Neither the nuclei nor the cytoplasmic membranes appeared to be stained by curcumin (white arrows). More attention paid to regions stained by curcumin together with white light diffraction images also suggested that curcumin is certainly associated with an intracellular compartment (Figure 1a).

We decided to investigate early curcumin staining by using time-resolved flow cytometry for two usual concentrations of curcumin, i.e., 5 µM and 20 µM (Figure 1b–d). Within 30 s, the curcumin was taken up by cells (Figure 1b,c) and reached a steady state that lasted for hours (see Figure 3h). Cell autofluorescence was low at 488 nm excitation and baseline gate (R1) mean fluorescence intensity was 3 ± 2, whereas curcumin-treated gate R2 (5 µM) and R3 (20 µM) were 46 ± 23 and 109 ± 25, respectively (Figure 1d).

### 3.2. Curcumin Is Not Localized on the Mitochondrial Network

Classic staining of the mitochondrial network with the mitochondrial potential probe tetramethylrhodamine methylester (TMRM, 40 nM) allowed the fine localization of mitochondria (Figure 2a). Yellow arrows indicate the mitochondrial compartment while white arrows show positions where no TMRM signal was detected (Figure 2a,b). Clearly, mitochondrial staining does not co-localize with more diffuse curcumin fluorescence that appeared often in the vicinity of mitochondria. The use of a microfluorometric approach allowed us to visualize roughly the cytoplasmic distribution of curcumin (5 µM) via its natural yellow/green fluorescence (Figure 2b). Microspectrofluorimetry enabled us to define within the same sample specific regions of the cell, i.e., yellow circles (1 and 2), where emission spectra were recorded (Figure 2c,d). The spectrum from region 1 was largely influenced by the presence of mitochondria and TMRM fluorescence whereas the curcumin signal was low, 4000 instead of 5250 (arbitrary units) (Figure 2c). The region 2 spectra had quite a low TMRM signature, corresponding to a few mitochondria, while the curcumin signal was higher than in region 1 (Figure 1c), excluding a direct link between curcumin and mitochondria (Figure 2d).

At this point, we observed that there was no curcumin in the extracellular medium (data not shown) and no significant fluorescence associated with the cytoplasmic membrane, nuclear membrane, nuclei or mitochondria. According to previous results and current microscopy, we hypothesized that diffuse curcumin fluorescence might be located in the ER network with a large proportion of the ER being in contact with the mitochondrial compartment.

With regard to the above considerations, we decided to concentrate our attention on the ER and the lysosomal compartment.

### 3.3. The Major Fraction of Curcumin Locates to the Endoplasmic Reticulum

By using the ER-specific fluorescent probe (ER-tracker™ red) (Figure 3b,d) and curcumin fluorescence (Figure 3a,c), we localized curcumin at the level of the ER network. Curcumin fluorescence (5 µM for 3 h) coincided widely with the ER-tracker™ red fluorescence (Figure 3b,c).

Additional experiments were conducted to follow the status of the ER compartment. ER swelling was evaluated since this is a common marker of stress, and the increased fluorescence of the ER-tracker™ red associated with the increased ER concentration is a coarse measurement of the ER lumen swelling (Figure 3e). The use of 4-phenylbutyric acid (PBA) as an ER stress inhibitor reduced this ER-tracker™ red fluorescence (Figure 3e) and minimized the ER swelling, as expected (not shown), showing that curcumin causes ER stress (with a UPR stress response) which is only partially reversed by PBA.

### 3.4. Curcumin Localization at the Endoplasmic Reticulum Causes a UPR Response Together with a Calcium Status Change

Increasing curcumin treatment inhibits the SERCA pump (sarco/endoplasmic reticulum Ca^2+^-ATPase), which is usually responsible for the transfer of calcium from the cytosol to the ER lumen at the expense of ATP (Figure 3f). On the other hand, the UPR stress response features an increase in the expression of C/EBP homologous protein (CHOP) and glucose-related protein (GRP78/BiP). We investigated by flow cytometry the expression of these proteins after 24-h treatment with various concentrations of curcumin. As the curcumin concentration increases, and especially for 20 and 50 µM, the content of CHOP and GRP78 increases, showing an active UPR response in Huh7 cells (Figure 3g). Calcium release experiments assessed by Fluo-4AM (cytosolic calcium) showed a fast increase in cytosolic calcium as soon as cells were treated with curcumin (Figure 3h). This increase is true in terms of intensity of staining, but also in terms of the number of cells involved, as 95% of the population showed this fast increase in cytosolic calcium. An initial pulse (before 30 min) was observed both for cytosolic calcium and curcumin, followed by a decrease in time to a level that was still above that of the untreated cell condition (Figure 3h). This permanently raised calcium level is certainly very important for the disruption of cell homeostasis.

### 3.5. A Second Fraction of Curcumin Is Lysosomal

Flow imaging was used to locate curcumin within the cell, notably in the lysosomal compartment (Figure 4). For this purpose, we assessed the intracellular distribution of the green fluorescence of curcumin and the red fluorescence of the Lysotracker red probe (Lysotracker Red). The histogram of curcumin fluorescence (97.4% of cells, Figure 4a) together with the histogram of the Lysotracker Red (97.4% of cells, Figure 4b) gave rise to a biparametric histogram with 87.1% of the cells being double stained (Figure 4c). Lysosome staining appeared heterogeneous but discrete, while curcumin fluorescence was much more widespread (Figure 4d). So, we performed a similarity analysis on the Amnis system to see whether the chosen cells (those presenting double staining) exhibited a similar colocalization of Lysotracker Red and curcumin in lysosomes (Figure 5). The histogram presented in Figure 5a shows that similarity scores under 1 correspond to a non-colocalized population (Figure 5b), while similarity scores over 2 concerned a significant colocalization signal observed in cells (Figure 5c). This population accounted for 14.8% and 23.2% of the total double-stained population cells, after incubation for 3 h with 20 µM and 50 µM, respectively (Figure 5a). This population went up to 95% after 50 µM for 24 h (Table 1). So, while the ER is stained by curcumin rapidly and homogeneously, the lysosomal compartment is also stained - but later on, and its full staining is attained with a longer time of incubation (and occurs more easily if the concentration of curcumin is raised) (Table 1). The duration of the lysosomal sequence of staining pointed to the ER as the main and primary intracellular target of curcumin.

### 3.6. Depicting Curcumin-Induced Autophagy and Apoptosis by Impedancemetry Combined with Flow Cytometry

The xCELLigence system allows the follow-up in real time of cell attachment, spreading, and proliferation, eventually leading to confluency, medium exhaustion, and cell death. Curcumin was added to the medium during the proliferation phase at different concentrations (Figure 6). At 50 µM curcumin (blue curve), the cell index dropped faster and sooner than under control conditions (black circles), indicating irreversible cell death (Figure 6a). At lower concentrations (25 µM; red curve), cells underwent a similar drop in cell index but progressed to a novel proliferation phase 24 h after curcumin addition. We hypothetized, that these observations could certainly be explained by intricated interactions between cell death and autophagic processes that may be temporarily linked by a cell cycle arrest. That is the reason why we subsequently investigated some characteristics of these cells at different timepoints along the proliferation curves using a multiparametric flow cytometric analysis.

The cell cycle study showed the expected profile for Huh-7 control cells; with the majority of cells in G_0_G_1_ (56%), and a large population in S phase (43%) that relied on the 8% proliferative cells in G_2_/M. Dead cells in culture were only 3% in a sub-G_0_G_1_ phase (Figure 6b). Cells treated with 50 µM curcumin exhibited a 41% dead cell population (sub-G0G1 cells), a low ratio of cells in S phase (20%), and an increase to 17% G_2_/M phase cells, suggesting cell cycle arrest (Figure 6d). Cells treated with lower amounts of curcumin exhibited an intermediate proportion of dead cells (24%), a large population in G_0_G_1_ (31%) and S phase (24%), but also a 21% population blocked in G_2_/M (Figure 6c). The secondary proliferation phase seen in Figure 6a showed a G_2_/M population decrease (15%) and only a 6% population of dead cells at the beginning of the curve (Figure 6e), ending up with a regular profile of cell growth at point 5 (Figure 6f). This toxicity associated with cell cycle arrest and a delay before a novel proliferating phase suggests a successful coping mechanism like autophagy.

Cell viability status along these timepoints was also investigated with YOPRO-1/PI staining membrane permeability assays (Figure 7). Untreated cells and those collected at timepoint #5—as cells that had escaped from death—exhibited a similar profile, with more than 90% of cells intact (Figure 7a,e). At timepoint 6, 96% of the cells were in the process of death and almost 60% of them were already necrotic (Figure 7f). At timepoint #2 (Figure 7b), we observed a 13.2% increase in necrotic cells, while at timepoint 4 there were still 24% of the necrotic cells (Figure 7d). More interestingly, the population of YOPRO-1-positive/PI-negative cells increased to 19.9% (Figure 7d); a consequence of the semi-permeability of the cytoplasmic membrane, which can, of course, indicate an apoptotic process, but also a transient state during autophagic processes. If autophagy is successful, it lifts the cell cycle blockade and gives a profile like timepoint #5 with a 90.3% viable population (Figure 7e).

We further investigated at these timepoints (Figure 6a) for evidence of apoptosis initiation, with a drop in mitochondrial membrane potential, production of superoxide anions, and an increase in cytoplasmic calcium levels (Table 2) [24,26,30]. Timepoint, metabolic events 6 (Table 2) clearly showed massive cell death, with only 4% of cells detectable with high ΔΨm, while other variables could not be measured as the cells were too damaged. Timepoints for metabolic events 2 and 3 showed a significant drop in ΔΨm associated with an increase in ROS and increased calcium levels (Table 2 metabolic events 1 and 2), consistent with previous results (Figure 6 and Figure 7). Except for ΔΨm, timepoint, metabolic events 2 displayed a lesser stress response than timepoint, metabolic event 3, which is consistent with the curcumin concentrations used. The population at high ΔΨm was still low at timepoint, metabolic events 4, but ROS and calcium were back to normal levels and cells were actively proliferating (Table 2 metabolic events 1). The interpretation of this profile could be that some cells had the ability to perform enough ROS detoxification and could return the transient calcium increase back to a functional state. This relies on a fine balance, where ROS induces autophagy but stays under a threshold that would cause alterations—e.g., lipid peroxidation and protein carbonylation—leading to cell death.

### 3.7. Number and Size of the Acidic Compartment as Evidence of Autolysosome Formation

To gather further evidence of curcumin-induced autophagy, we analyzed variation in the intracellular acidic compartment of treated cells (20 µM curcumin for 16 h) by flow imaging (Figure 8) and flow cytometry (Figure 9). Bright field images clearly showed an increase in subcellular structures in curcumin-treated cells compared to controls (Figure 8a–c). As previously reported, the acridine orange probe (AO) accumulates in vesicles and stains the acidic compartment in red (AO red) [31]. A clear increase in AO red staining was observed in curcumin-treated cells, both in terms of intensity and in terms of numbers of punctae throughout the cells (Figure 8). Moreover, AO red fluorescence doubled in intensity and reached a maximum after only one hour of 20 µM curcumin (Figure 9a). This increase in size and number of the acidic compartments suggested an increase in both autophagic vesicles and lysosomes followed by their fusion (Figure 8c). The size of control isolated lysosomes reached 0.9 ± 0.8 µM, while alleged autolysosomes reached 1.95 ± 1.1 µm (Figure 9b). The treatment of the cells with curcumin (Figure 8c) promotes a huge increase in acidic vesicles and autophagic vacuoles. Electron microscopy analyses of curcumin-treated cells confirmed the classic features of stress like swollen mitochondria with translucid cristae devoid of membranes, surrounded by ER also swollen with extensive lumen (Figure 9c,d). Some stress fibers (StF) were also observed as where the lysosomes (Lys) and large autophagosomes (Figure 9c–f). Curcumin rapidly mobilized the autophagic capacity of the cells in order to discard unwanted and dysfunctional cellular compartments.

## 4. Discussion

Many studies have investigated the use of curcumin for therapeutic purposes, though few have quantified curcumin uptake or analyzed its intracellular localization. Despite its poor bioavailability, a fraction of curcumin is able to cross the plasma membrane and accumulates in the cell within seconds after its addition to the extracellular medium. The intracellular level of curcumin is linearly related to the extracellular content and the concentration ratio is 20 to 1 [8]. By combining imaging and flow cytometry, we were able to follow the intracellular curcumin staining in Huh-7 cells. Flow imaging revealed that almost all cells were stained. Despite diffuse intracellular staining, curcumin did not localize at the cytoplasmic membrane or at the nuclear membrane (Figure 1a and Figure 4d). This staining and the intracellular accumulation were not related to any destabilization of the plasma membrane, since 97.5% of Huh-7 cells were intact in these conditions [8].

### 4.1. Curcumin Does Not Localize at the Mitochondria

Previous publications show that curcumin is involved in mitochondrial destabilization [8,32] and induces mitochondrial biogenesis [33]. Hypothetically, if curcumin treatment increases the ratio of Bax/Bcl-2 or results in a leaky mitochondrial membrane, curcumin could be linked to mitochondrial destabilization, apoptosis, and mitophagy [8,18,32,33,34]. However, even if this hypothesis it correct, this does not need a direct interaction between curcumin and the mitochondrial membrane.

Our microspectrofluorimetry assays indicate that curcumin does not stain the mitochondrial compartment itself, but rather its vicinity (Figure 2). So, it can be supposed that the ER compartment, which is known to be tightly link to the mitochondrial network [35,36,37] is one of the main target of curcumin. The new concept of “contactology” has been applied to the study of the physical and functional bridge between the ER and mitochondria [35]. The main proof that mitochondria are not the primary target of curcumin is that when needed curcumin targeting to mitochondria is achieved by curcumin grafting with tetraphenyl-phosphonium (TPP^+^), which will favor the mitochondrial localization of the construct mediated by the cationic charges and enhanced cytotoxicity [38].

One can also hypothesize that curcumin interacts with transcription factors likely to act on efficient coordinators of mitochondrial biogenesis and perhaps of autophagy, without the need for any direct interaction with mitochondria [33].

### 4.2. Curcumin-Induced Endoplasmic Reticulum Alterations and Its Consequences

Confocal images of curcumin autofluorescence and ER-Tracker red clearly identified the ER as the main intracellular target of curcumin (Figure 1a–d). This ER localization is concomitant with a discrete increase in ER lumen size (Figure 2e) that may be related to the UPR response (UPR stress). ER stress is a very common feature of many physiological and pathological conditions affecting protein secretory pathway functions. The UPR is an adaptive response to ER stress. When disturbed, ER reacts to the burden of unfolded proteins in its lumen (ER stress) by activating intracellular signal transduction pathways, collectively termed UPR.

This UPR response is activated to lower the misfolded or unfolded protein content. When ER stress becomes intense, the UPR response shifts to autophagic and pro-apoptotic signaling capabilities ending in cell death [39,40]. To ascertain fidelity to correct protein folding, cells regulate protein-folding capacity in the ER according to need.

The underlying mechanism of lumen swelling we described is partially dependent on 4-PBA that is commonly used to alleviate ER stress (Figure 2e) [41]. 4-PBA is supposed to interact with the hydrophobic domains of misfolded proteins, thereby preventing their aggregation. Nevertheless, inhibition of the lumen size increase is limited, suggesting that other mechanisms could be at work and that simple curcumin insertion into the membrane inducing ER permeabilization could be enough as a complementary mechanism.

Beyond its primarily ER localization, curcumin inhibits Ca^2+^-ATPase (SERCA), thereby inducing an increase in cytoplasmic calcium (Figure 3f). These results support previous reports showing that curcumin inhibits the SERCA Ca^2+^-pump, via the inhibition of its Ca^2+^-dependent ATPase activity, with an IC_50_ value between 7 and 15 µM (external concentration) [42,43]. It appears that curcumin acts through inhibition of the Ca^2+^-dependent ATPase and conformational changes (curcumin stabilizes the E1 conformational state), which fully abolishes ATP binding. This observation could be directly linked to intracellular calcium increases (Figure 3h). Furthermore, ER lumen swelling was associated with an increase in the number of cells exhibiting a high calcium level (Figure 3h). The sequence of the calcium fluorescence increase induced by curcumin looks like a pulse of calcium, followed by a decrease in calcium-associated fluorescence to a level that is somewhat higher than before the addition of curcumin. The pulse duration lasted almost two hours.

There is also increased expression of CHOP (CCAAT-enhancer-binding protein homologous) [44,45] and of the 78-kDa glucose-regulated protein, GPR78 [46] (Figure 3g)—an ER molecular chaperone also referred to as BiP and generally associated with the UPR response. Indeed, GRP78/BiP is required for stress-induced autophagy [47].

### 4.3. Curcumin and Secondary Lysosomes Stained

When its concentration increased, curcumin also stained a portion of the lysosomal compartment that is growing (Figure 4 and Figure 5 and Table 1). In this context, it is well known that curcumin is able to bind directly to TFEB (transcription factor EB [48], a master regulator of autophagy and lysosomal biogenesis. The curcumin functional groups suitable for interaction (direct binding) with other macromolecules include the α,β-unsaturated β-diketone moiety, carbonyl and enolic groups of the β-diketone moiety, methoxy and phenolic hydroxyl groups, and the phenyl rings [4].

The current literature shows that the subcellular localization of TFEB is lysosomal and that its delocalization is very sensitive to lysosomal stress [45,46]. As 10–20 µM curcumin activates lysosomal destabilization and induces autophagy, lysosomal stress secondarily activated by curcumin may also activate lysosomal biogenesis and autophagy.

### 4.4. Successful Curcumin-Induced Autophagy Detected by Impedancemetry Coupled to Multiparametric Flow Cytometry Analysis

The xCELLigence impedancemetry set-up yielded unexpected results (Figure 6a). We observed a classic proliferation curve with untreated cells, i.e., a proliferation phase until reaching a plateau, before cells started to condense (lower IC) and detach (no IC) as they underwent cell death due to nutrient deprivation in the wells. The use of two concentrations of curcumin, i.e., 20 and 50 µM, gave us two distinct curves. The 50 µM curve is typical of the induction of cell death (Figure 6a), whereas the 20 µM curve is very surprising: after a proliferation slope similar to the one at 50 µM curcumin during the first 16 h following curcumin administration, the curve changed shape and the IC stopped decreasing, and then increased again in a second proliferation curve (Figure 6a point 4).

The cell cycle analysis provided a more precise idea of what happened to the cells. At 20 µM curcumin, the cells clearly escaped cell death and one may suspect successful autophagy allowing the cells to proliferate again. Cells (Figure 6c) first exhibited a cell cycle similar to that of dying cells (50 µM curcumin treatment) (Figure 6d), except that dead cells in sub-G_0_G_1_ were less abundant. Most importantly, cells in Figure 6e,f exhibited the characteristics of cells re-entering the cell cycle before completely recovering and being engaged in a new proliferation step.

Many publications assume that curcumin inhibits the proliferation of several types of cancer cells [49]. The arrest in cell-cycle progression is generally positioned at the G2/M phase of the cell cycle [50], which is also the case with the curcumin-treated MCF-7 cells [51]. The curcumin arrest at the G2/M phase might be explained by the fact that curcumin was found to perturb microtubule-kinetochore attachment and also activated the mitotic checkpoint, resulting in delayed mitosis [52,53,54]. It has also been demonstrated that treatment of MCF-7 cells with higher curcumin concentrations (over 20 µM) increased the number of cells in the sub-G_0_G_1_ phase related to increased apoptosis [49,53,54].

The monitoring of 50 µM curcumin-treated cells reflects the important loss of cell viability associated with cells undergoing classic cell death (Figure 7c,f). The cells (20 µM treatment) that entered a new proliferation step after transition (Figure 7b,d,e) never presented a similar decrease in cell viability. There were 58.1% of viable cells plus as few as 19.9% of them going through a permeability state (non-permanent loss of viability) that is reversible. Indeed, 78% (58.1% + 19.9%) of cells maintained more or less their plasma membrane in an almost impermeable state (these cells did not become PI-positive) (Figure 7d).

Moreover, if we follow the cells along the impedance curve during their proliferation state (IC increase) and/or their curve of IC decrease, and measure their characteristics in terms of early events associated with cell death induction [24,26], we can draw a clear picture of their behavior. Usually, early cell death is detected through a drop in mitochondrial potential (ΔΨm) associated with the immediate generation of ROS (superoxide anions and hydrogen peroxides) associated with an increase in intracellular calcium [25,51]. This is the situation recorded for the 50 µM curcumin-treated cells (Table 2), whereas it is not the same when cells are treated with 20 µM curcumin. These cells undergo some change in ΔΨm, but the number of cells with a low ΔΨm is not a majority (38%) (Table 2 metabolic events 4), and the cells exhibited increased ROS production and calcium when compared with cells undergoing apoptosis, where 50% of cells showed a drop in ΔΨm. More significant is the fact that the proportion of cells with ROS and calcium is approximately the same (≥ 45%) (Table 2 metabolic events 4). This means that curcumin′s induction of the cellular antioxidant system and/or efficient mitophagy associated with the restoration of ΔΨm may push almost 50% of cells to go through the cell cycle and escape cell death.

It is quite interesting to hypothesize that the alteration of lysosomes and activation/delocalization of TFEB may lead to mitochondrial biogenesis [55]. In parallel, this is also associated with an increased level of mitophagy and overall this results in a restored quality of mitochondrial functions [55]. On the other hand, the anti-inflammatory action of curcumin [56,57,58] has two aspects: the first is mediated through its ability to inhibit cyclooxygenase-2 (COX-2), lipoxygenase (LOX), and inducible nitric oxide synthase (iNOS) [59,60], and the second is mediated through the ability of curcumin to activate directly antioxidant enzymes, i.e., catalase (CAT), superoxide dismutase (SOD), and glutathione peroxidase (GSH-PX) [48]. Low- and medium-concentration curcumin decreased malondialdehyde (MDA) and ROS levels and nfr2 KEAP-1 induction [58].

Therefore, our results suggest that curcumin, which drives successful autophagy (mitophagy), not only protects cells (after a first step of toxicity-induced events) but also reverses the mitochondrial damage and dysfunction induced by oxidative stress.

### 4.5. Curcumin at Low Concentration Exacerbates Autophagy (Mitophagy)

Image cytometry confirmed the large increase in the acidic compartment when stained by AO (Figure 8). It is clear that AO helps to monitor the increase in synthesis of lysosomes, but also the generation of autophagic vesicles (Figure 8a,b). This is correlated with previous reports concerning the use of AO as a reporter of lysosomal synthesis and autophagy [31,59] Moreover, these data confirm our previous reports that curcumin induces autophagy with a clear LC3-II increase [8]. These events unfolded in parallel to lysosomal compartment amplification, as described previously [8]. It is clear in our case that curcumin induces an increase in the number and size of the lysosomes (Figure 9b), and this fits very well with the late fusion of lysosomes to form autolysosomes. It has been demonstrated that curcumin at moderate concentration induces lysosome proliferation through TFEB delocalization, which is a key feature of autophagy induction [45,46,48,60].

Evidence in previous publications clearly shows that curcumin is rapidly internalized within cells, even if the extra-/intracellular concentration ratio is 20 to 1. So, working within 0 to 50 µM curcumin implies that the intracellular curcumin concentration is in the range of 0 to 2.5 µM. Internalized curcumin localizes first at the ER and second at the lysosomes (Lys). In both cases, curcumin destabilizes these compartments and induces a UPR in the ER and a lysosomal stress response.

Originally, we used impedancemetry to investigate the fact that curcumin can induce both autophagy and cell death. Thie flow cytometric analysis of the cells coming from xCELLigence favored our capability to depict sequential events produced by curcumin. At 25 µM, curcumin first induced cells to enter cell death processes that rely on apoptosis and necrosis, but also started autophagic processes that eventually—if successful—allowed cells to recover, escape the G_2_/M blockade, and proliferate again (Figure 6). In contrast, if curcumin is used at higher concentrations, the insults at both the ER and lysosomes are too great to be overcome by coping mechanisms like autophagy, and so full type II cell death signaling occurs, with no possible return. These results confirm that the mechanisms of curcumin operate at different levels of the cell, but also at different times and intensities, as summarized in Figure 10, a paradigm at first partly described with previous data [8,18].

## Figures and Tables

**Figure 1 cells-09-00361-f001:**
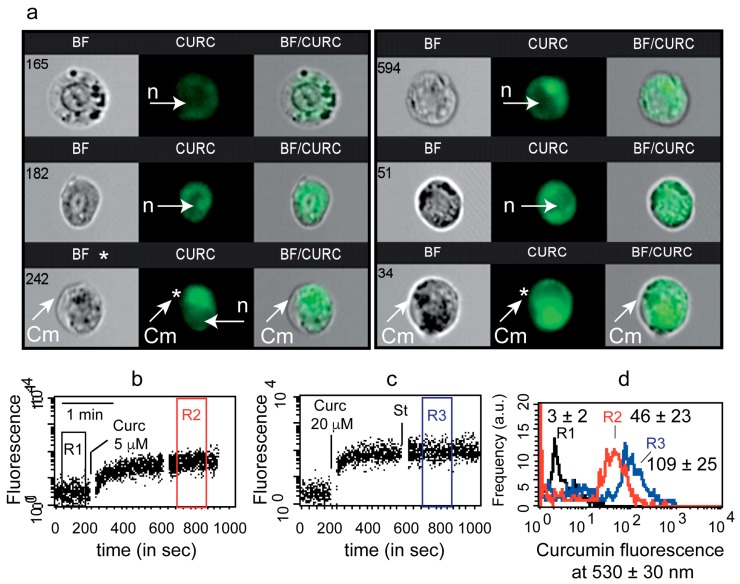
Qualitative analysis of curcumin staining of cells. (**a**)—Image flow cytometry of curcumin uptake by Huh7 cells. White arrows indicate either nuclei (n) or cytoplasmic membrane (Cm). (**b**,**c**)—Cell fluorescence over time before and after 5 µM curcumin (**b**) or 20 µM curcumin (**c**) was added to the tube. (**d**)—Curcumin fluorescence intensity of different regions of previous kinetic measurements. The mean fluorescence values of curcumin in R2 and R3 are directly noted on the histograms.

**Figure 2 cells-09-00361-f002:**
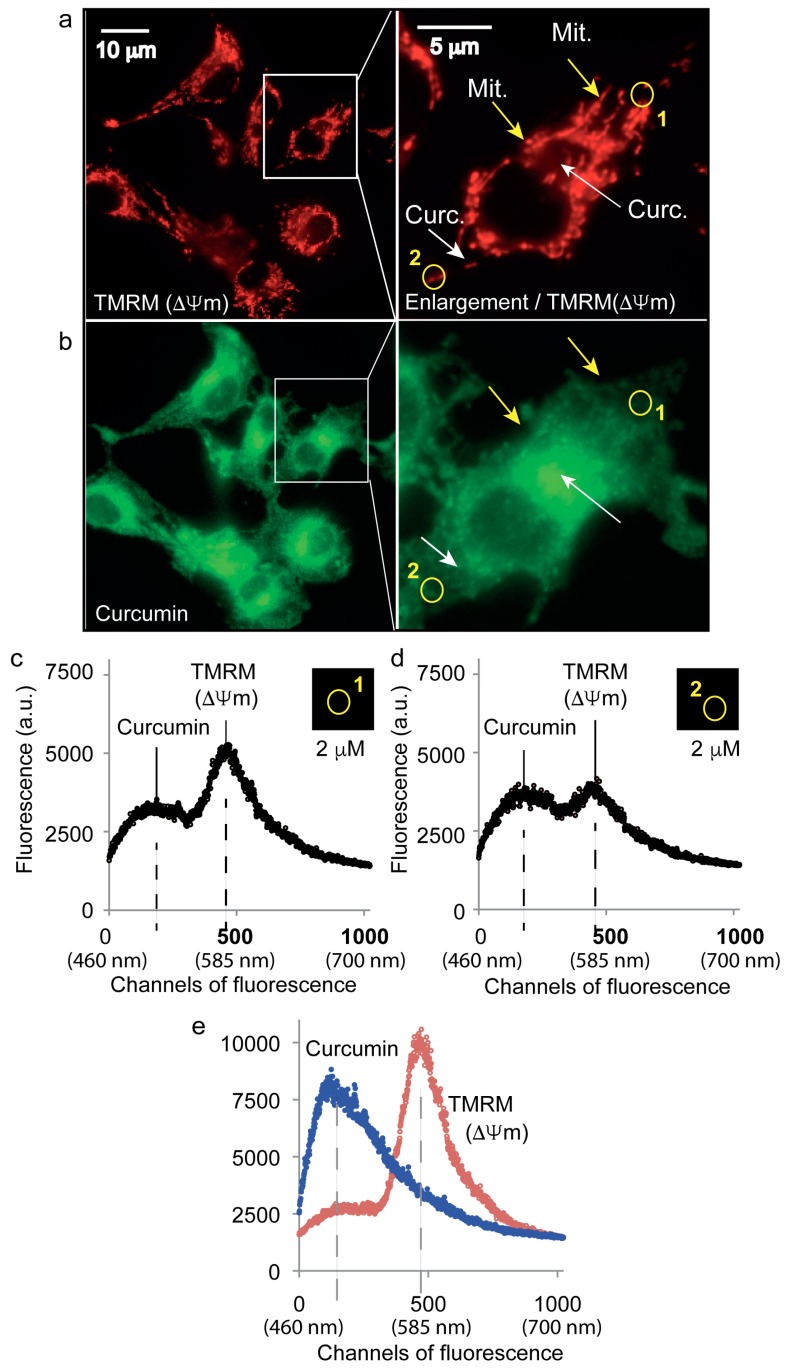
Intracellular curcumin is not found in the mitochondrial network. (**a**)—Fluorescence microscopy of the mitochondrial specific dye, TMRM, at 585 ± 42 nm after excitation at 488 nm. (**b**)—luorescence microscopy of curcumin at 530 ± 30 nm after excitation at 488 nm. White arrows indicate regions with high curcumin fluorescence, yellow arrows indicate regions with a high density of mitochondria. (**c**,**d**)—Microspectrofluorimetric spectra at selected regions 1 and 2 (shown in **a**,**b**). Panels (**c**,**d**) are related to the areas named 1 and 2 in (**a**,**b**). The relative values of curcumin and TMRM in areas 1 and 2 are evaluated at the points of maximum emission of curcumin (530 nm) and TMRM (585 nm). (**e**)—Microspectrofluorimetric spectra of curcumin or TMRM alone showed peaks around 530 nm and 585, respectively, in their free forms.

**Figure 3 cells-09-00361-f003:**
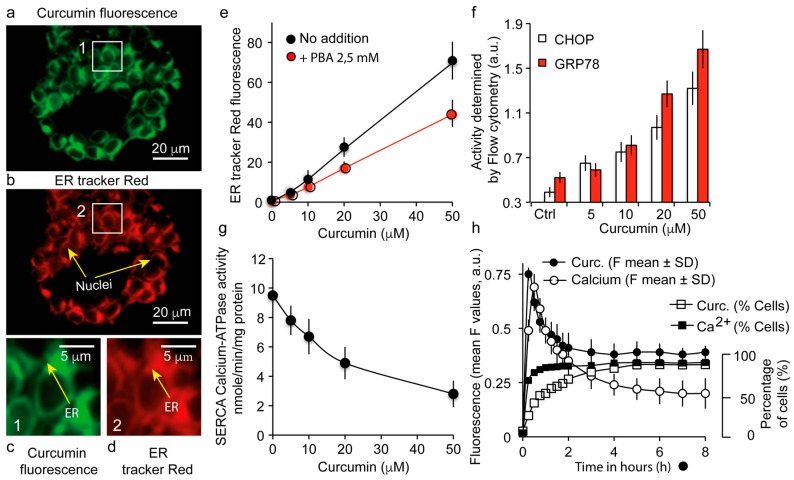
Localization of intracellular curcumin at the endoplasmic reticulum (ER) and associated signaling events. (**a**,**b**)—Confocal analysis of the double staining with curcumin (in green) and ER-tracker red, (in red). Arrows indicate the ER compartment; (**c**,**d**) panels are enlargements of the two boxes numbered 1 and 2 of pictures (**a**,**b**). (**e**)—Volume of the ER compartment assayed by the ER tracker red fluorescence in flow cytometry with different concentrations of curcumin with or without the ER stress inhibitor 4-phenylbutyric acid (PBA); (**f**)—Ca^2+^-ATPase measurements illustrating SERCA inhibition by different curcumin concentrations; (**g**)—Histogram representation of CHOP and GRP78 measurements following incubation with different curcumin concentrations illustrating the UPR response; (**h**)—Flow cytometry analysis of curcumin and calcium contents over time (in hours), by curcumin autofluorescence and Fluo4-AM, respectively.

**Figure 4 cells-09-00361-f004:**
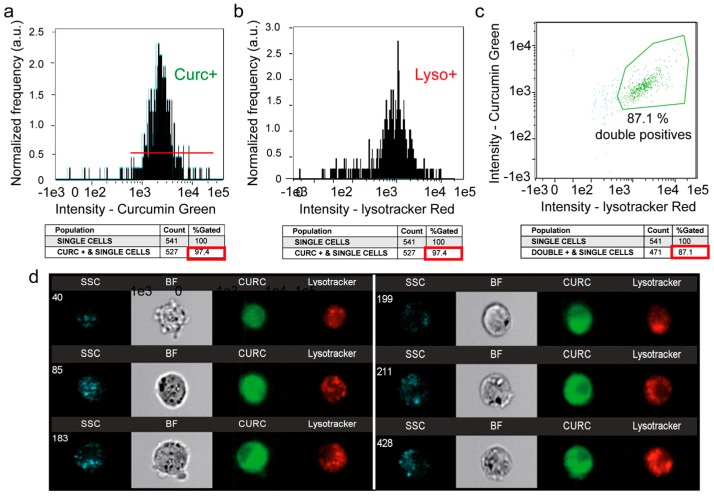
Image cytometry (Amnis) of the double staining: LysoTracker Red and curcumin. (**a**) Curcumin green fluorescence for 24-h incubation with 20 µM curcumin. (**b**) LysoTracker Red fluorescence analysis for 10-min incubation with 100 nM LysoTracker Red. (**c**) Fluorescence histogram of curcumin and LysoTracker green fluorescence. (**d**) A selection of 6 cells analyzed with Amnis where it is possible to see the widespread intracellular staining of 5 µM curcumin for 3 h and the punctate staining with LysoTracker Red.

**Figure 5 cells-09-00361-f005:**
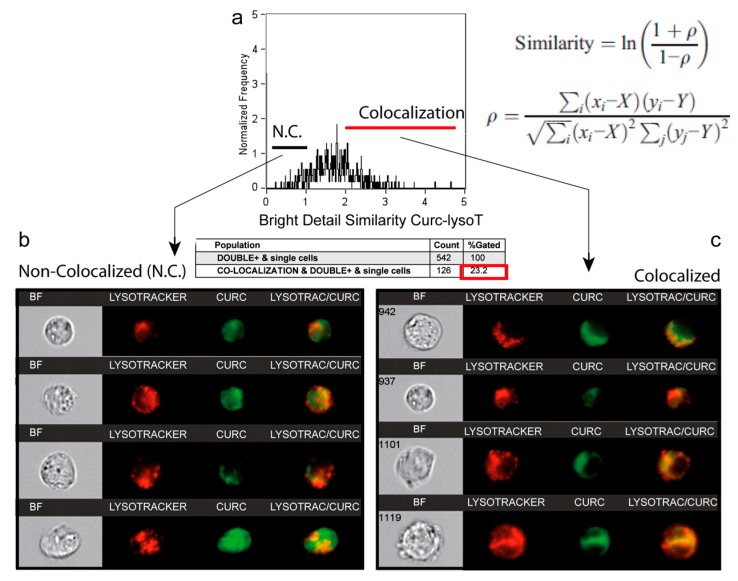
Similarity analysis for correlation between curcumin fluorescence and LysoTracker Red. (**a**) Analysis of the colocalization of curcumin and LysoTracker red (lysosomes) by using the Similarity Score included in IDEAS 6.0 software™ (Amnis). This score, a log-transformed Pearson’s correlation coefficient between the pixels of two image pairs, provides a measure of the degree of co-localization by measuring the pixel intensity correlation between the curcumin and LysoTracker images. Analysis performed on 542 cells. Cells that were permeable to TO-PRO-3 iodide and/or debris were excluded from the analysis together with cellular aggregates. This corresponds to 5 μM curcumin and 3-h incubation plus 10-min staining at 37 °C with 100 nM LysoTracker Green. (**b**) Selection of some images that correspond to cells (from the histogram in a) where there is no strict correlation between curcumin and LysoTracker Red. (**c**) Selection of some images corresponding to the cells presenting full colocalization of the two probes from the ahistogram in (**a**).

**Figure 6 cells-09-00361-f006:**
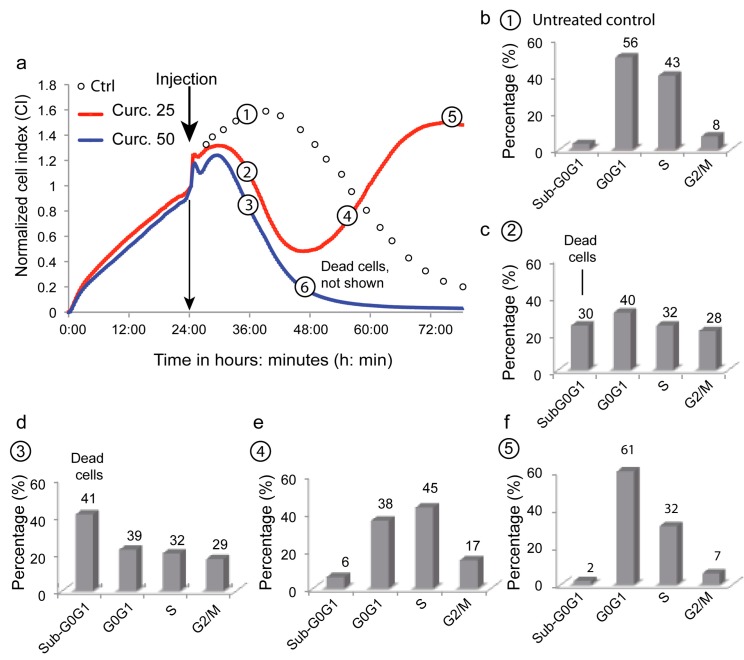
Combined xCELLigence and flow cytometry analysis of the Huh-7 cell cycle at two curcumine concentrations (i.e., 20 and 50 μM). Cells were treated with 20 µM (red line) or 50 µM (blue line) curcumin. The control was imaged with circles after the injection position since it was correlated with either the red or blue line in the first 24 h (we did not want superimposition of the control line with the two others). Circled numbers (1–6) indicate the times when cells (4 wells each time) were taken up from the xCELLigence set-up to be stained and analyzed by flow cytometry of the cell cycle. The percentage of cells in each part of the cell cycle (G_0_G_1_, S, G_2_/M) is indicated, and the sub-G_0_G_1_ indicates dead cells (DNA concentration and less staining with PI). Cells are from positions 1–6 along the curve and are labeled (**a**–**f**).

**Figure 7 cells-09-00361-f007:**
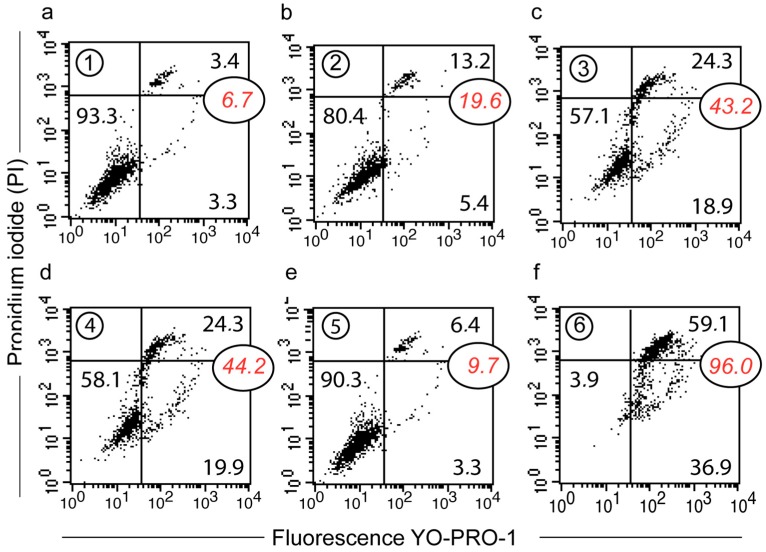
Detailed analysis (**a**–**f**) of the change in plasma membrane permeability of Huh-7 cells (using the xCELLigence device). YOPRO-1/PI staining was used and three cell populations could be distinguished: YO-PRO-1^−^/PI^−^ for viable cells, YO-PRO-1^+^/PI^−^ or YO-PRO-1^+^/PI^intermediate^ for apoptotic cells and YO-PRO-1^+^/PI^+++^ for necrotic cells. The histograms are representatives of nine different experiments realized with the same flow cytometer setting.

**Figure 8 cells-09-00361-f008:**
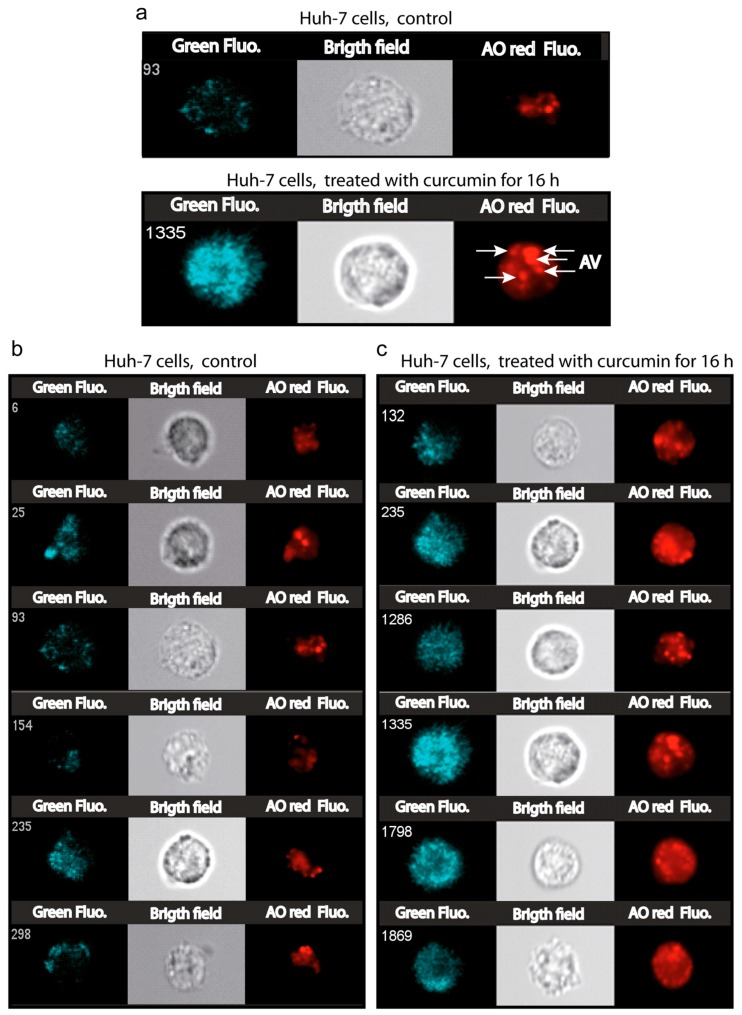
Image cytometry of the acidic compartment of Huh-7 cells treated with 5 μM curcumin for 16 h. The measurements were done with an Amnis ImageStream100 (Amnis, Merck Millipore) imaging flow cytometer. (**a**)—The images presented are dark field (SSC, Ch01), bright field (Ch02 gray) and the red channel (Ch04) for the red nonyl acridine fluorescence from the NAO red aggregates in a low pH environment. Control Huh-7 cells with lysosome staining essentially and cells treated with curcumin where staining is accentuated in lysosomes, but also in acidic autophagic vacuoles. (**b**)—same as in a, exhibiting six distinct cell staining experiments and (**c**) for 16-h treatment with 5 μM curcumin.

**Figure 9 cells-09-00361-f009:**
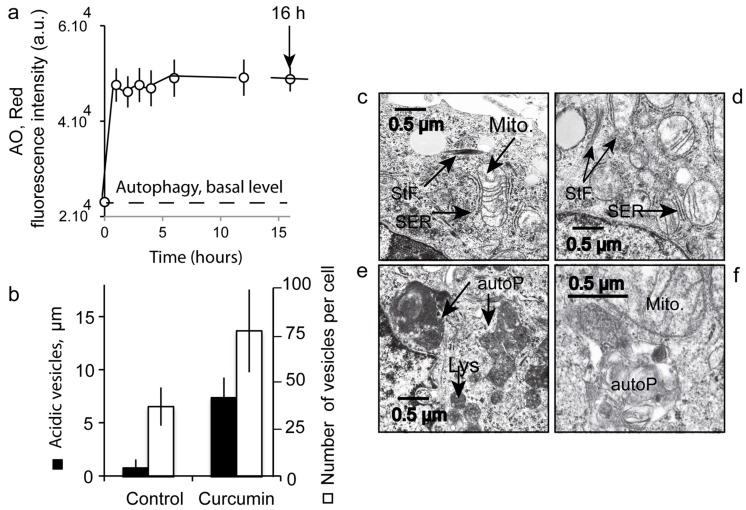
Detection of the acidic vesicles by fluorescence and/or electron microscopy. (**a**)—Flow cytometry analysis of acridine orange (AO) staining of the acidic vesicles in samples treated with curcumin for 16 h. The histogram presents the fast increase in AO internalization in low pH vesicles. (**b**)—Image cytometry measurements presented in Figure 8. The sizes of the vesicles are given in µm and the number of vesicles is indicated. (**c**,**d**)—Electron microscopy picture of a sample of Huh-7 cells treated with 20 µM curcumin for 16 h. In (**a**), we can see essentially mitochondria (Mito.) with swollen ER (SER) and stress fibers (StF). In (**b**), essentially round, translucent mitochondria defective in cristae membranes with swollen ER (SER) and stress fibers (StF): in (**e**,**f**), essentially lysosomes (Lys) and autophagolysosomes (AutoP). With only one mitochondrion can be seen (Mito) in (**f**).

**Figure 10 cells-09-00361-f010:**
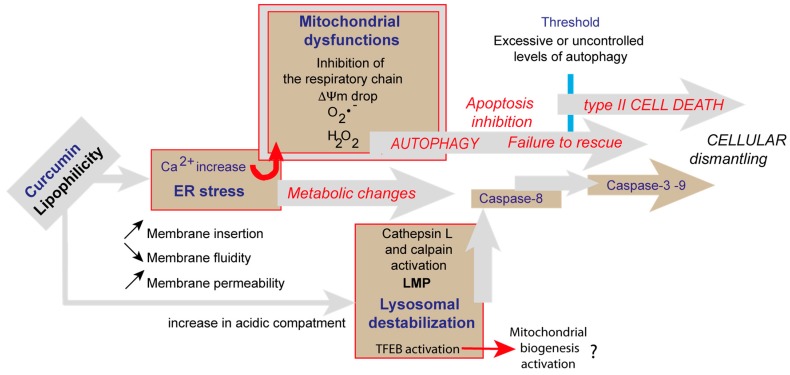
Schematic representation of the various events involved in curcumin-induced signaling. Lipophilic curcumin penetrates the cells and interacts with some of the membrane compartments. Curcumin primarily provokes ER stress and secondarily lysosomal membrane destabilization. Calcium released from the ER enters the mitochondria and alters the electron transport chain, resulting in a ΔΨm drop and superoxide generation. Altered mitochondria are taken up by mitophagy. The lysosomal pathway also converges towards mitochondria since calpain and cathepsin are likely to activate caspase-8 and cleave Bid (giving rise to tBid and mitochondrial membrane permeabilization). Huge autophagic processes are induced at low curcumin concentration in parallel to cell death events until beclin-1 is cleaved and cell death dominates. Interestingly, bafilomycin A1 inhibition of autophagy leads to more pronounced cell death.

**Table 1 cells-09-00361-t001:** Interaction of curcumin with the lysosome as a function of its concentration and incubation time. The percentage (%) of cells that exhibit both curcumin and LysoTracker fluorescence is given, as is the percentage (%) of cells that exhibit strict colocalization, meaning that curcumin binds to lysosomes.

**Curcumin Incubation 3 h**	**5 μM**	**20 μM**	**50 μM**
True colocalisation of curcumin and lysotracker red cells (% of cells)	1.95	14.8	23.4
Double curcumin and lysotracker red Staining (% of cells)	63.6	66.7	95.1
**Curcumin Incubation 24 h**	**5 μM**	**20 μM**	**50 μM**
True colocalisation of curcumin and lysotracker red cells (% of cells)	67.8	87.6	97.8
Double curcumin and lysotracker red Staining (% of cells)	92.2	95.3	96.8

**Table 2 cells-09-00361-t002:** Flow cytometry analysis of curcumin-treated cells from the xCELLigence system. The percentage (%) provided corresponds to the cells characterized as follows: high mitochondrial membrane potential (viable cells), cells presenting an increase in either superoxide anions or hydroperoxide, and cells with increased cytoplasmic calcium.

Metabolic Events (% of Cells)	1	2	3	4	5	6
ΔΨm high	95 ± 2	58 ± 4	50 ± 5	62 ± 5	97 ± 2	4 ± 1
Superoxide anions	4 ± 3	36 ± 6	49 ± 5	12 ± 2	2 ± 2	NA
Peroxidase activity	3 ± 4	35 ± 7	48 ± 6	10 ± 3	3 ± 3	NA
Calcium rise	1 ± 1	30 ± 8	45 ± 7	5 ± 3	1 ± 1	NA

Xcell: xCELLigence and the numbering Xcell 1 to Xcell 6 is the same number contained into circle (1 to 6) that are a along the impedance (IC) curves. NA: not analyzed due to the very low number of viable cells and the high amount of cellular debris.

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
