# Peer review of "Highlighting Curcumin-Induced Crosstalk between Autophagy and Apoptosis as Supported by Its Specific Subcellular Localization"

_cells, 2020, doi:10.3390/cells9020361_

Round 1
Reviewer 1 Report
Curcumin, a main compound identified in the Curcuma longa (turmeric), has brought many interests in the last few years due to its activities as antioxidant, anti-inflammatory agent and anti-cancer agent. Although curcumin has been recognized as a compound with a relatively safe profile as it has already been regarded for its therapeutic effects in traditional herb for several hundred years, its mechanism(s) of action(s) remains unclear.
In this manuscript, Oyanguren et al. investigated the subcellular localization of curcumin upon entering cells and concluded that it was mainly enriched at ER which may correlate with its activity for inducing the activation of endoplasmic reticulum (ER) stress and subsequent autophagy/apoptosis signaling process. However, the methodologies used by the authors were not comprehensive which may provide weak evidence for supporting many of their conclusions. Furthermore, much improvements in their writing including formatting and typos etc are needed.
Major points:
Page 5, Material and Methods 2.10, detection of CHOP and GRP78 protein by FACS cannot be used to measure the activities of these two proteins.
Page 6, last paragraph, “….and reached a steady state that lasted for hours (Figure 1b, c), the description is not appropriate as 1000 sec only equal to 16 min and 40 sec,
Page 8, Figure 2, TMRM is a cell permeable dye which is used only to label active mitochondria, since curcumin would cause mitochondrial depolarization, it is not suitable to use that as mitochondrial marker.
Page 8, first paragraph, lane 5, “…hypothesized that diffuse curcumin fluorescence is located in the ER network”, this hypothesis makes no sense as the fluorescence signal of ER should be tubular-network shape instead of diffuse everywhere in the cytosol.
Page 9, Figure 3c, the staining pattern of curcumin is different from what has been shown in Figure 2b, it is unconvincing that curcumin signal colocalized with ER but not mitochondria unless the authors showed the figures side by side.
Page 10, Figure 4, “Flow imaging was used to locate curcumin within cells, notably in the lysosomal compartment”, the interpretation of result using this method was unconvincing that curcumin is indeed colocalized with lysosome. The authors should conduct immunofluorescence confocal imaging analysis together with ER and mitochondrial marker.
Page 12, 3.6, first paragraph, “…these observations could be explained by cell death only or associated with autophagic processes”, In the absent of evidence suggesting autophagy signaling is involved, their conclusion is unconvincing.
Page 13, lane 4, “…60% of them were already necrotic (Figure 7f)”, double positive of PI and YOPRO-1 suggested the cell death could be either in the late stage of apoptosis or necrosis.
Page 15, Figure 9b, “while alleged autolysosomes reached 7.85 ± 1.9 mm”, that size mentioned by the author suggests that the autolysosomes are almost as big as nucleus, which is unlikely.
Other additional points:
Page 2, Introduction, first paragraph, lane 5, “C21H20O6” should be “C21H20O6”.
Page 5, 2.6, second paragraph, lane 1, “37mC” should be “37OC”.
Page 5, 2.10, lane 1 and 2, the description is obviously copied from other sources as there were no “Caco-2 cells” and “cocktails of pollutants” had been used in their entire study.
Page 7, lane 1 to 5, the description of Figure 2b is in front of Figure 2a, the text description and the figure layout should be coherent.
Page 8, Figure 2e, there’s no description of this panel until the final section of Discussion.
Page 8, Figure 3b, there’s no description of 3b in the main text.
Page 8, Figure 3b, the scale bar is mislabeled.
Page 8, Figure legend of Figure 3, lane 8, the description is obviously copied from other sources as there were no “p38” be measured in their entire study.
Reviewer 2 Report
The paper “Highlighting curcumin-induced crosstalk between autophagy and apoptosis: A biochemical approach coupling impedancemetry, imaging, and flow cytometry” by Francisco J. Sala de Oyanguren et al, is an interesting work describing the ability of different concentrations of Curcumin to induce a selective response in Human hepatoma-derived Huh-7 cells, shifting beetween autophagic and apoptotic pathways.
Furthermore the paper highlights the main interactions of curcumin with specific organelles, such as mitochondria, ER and lysosomes. The paper is well organized with the multiparametric approach described in the title, however it is possible to identify some major and minor points.
Major points:
The title should be modified, because it does not describe the selectivity and the timing of curcumin permeabilization in regards of the various organelles In the Discussion the part referring to other papers (from the same authors or different authors) is slightly confusing with the findings of the paper to revise, please revise in particular 4.1 (first 5 lines) and 4.2 (first 7 lines). Since Curcumin can also be excited at 488 nm, with a lower fluorescent yield emission in the 500-530 nm range, please provide (to reviewer) some hystograms illustrating FCM evaluation of fluo-4/AMM, Calcein-AM, 2,7-dichlorodihydrofluorescein diacetate (DCFH-DA), and controls adopted for YO-PRO-1/PI evaluation.
Minor points:
Pag 5 of 26, par 2.6 change 7-AAD instead 7ADD Please modify table 2 and composed it in the same way of table 1 (same titles), indicating, in a more visible manner, that data are from xCELLigence In figure 8 AO data are shown but NAO compares in the caption, please modify Other typos or english mistakes are scattered in the text The Discussion should be revised in the overall organization to render more homogeneous and armonised the text, as in the introduction The Abstract should be take more into account the interesting and new results regarding selectivity and the timing of curcumin permeabilization in regards of the various organelles.Author Response
Please see the attachment.

Round 2
Reviewer 1 Report
The authors had made sufficient improvements to the manuscript and the current version is acceptable for publication.